# Using Android Smartphones to Collect Precise Measures of Reaction Times to Multisensory Stimuli

**DOI:** 10.3390/s25196072

**Published:** 2025-10-02

**Authors:** Ulysse Roussel, Emmanuel Fléty, Carlos Agon, Isabelle Viaud-Delmon, Marine Taffou

**Affiliations:** 1Institut de Recherche Biomédicale des Armées, 91220 Brétigny-sur-Orge, France; ulysse.roussel@ircam.fr (U.R.); marine.taffou@def.gouv.fr (M.T.); 2STMS, Ircam, CNRS, Sorbonne Université, Ministère de la Culture, 75004 Paris, France; emmanuel.flety@ircam.fr (E.F.); agonc@ircam.fr (C.A.)

**Keywords:** remote assessment, mobile application, timing precision, automated hardware testing, reaction time, audio–tactile integration, peripersonal space

## Abstract

Multisensory behavioral research is increasingly aiming to move beyond traditional laboratories and into real-world settings. Smartphones offer a promising platform for this purpose, but their use in psychophysical experiments requires rigorous validation of their ability to precisely present multisensory stimuli and record reaction times (RTs). To date, no study has systematically assessed the feasibility of conducting RT-based multisensory paradigms on smartphones. In this study, we developed a reproducible validation method to quantify smartphones’ temporal precision in synchronized auditory–tactile stimulus delivery and RT logging. Applying this method to five Android devices, we identified two with sufficient precision. We also introduced a technique to enhance RT measurement by combining touchscreen and accelerometer data, effectively doubling the measure resolution—from 8.33 ms (limited by a 120 Hz refresh rate) to 4 ms. Using a top-performing device identified through our validation, we conducted an audio–tactile RT experiment with 20 healthy participants. Looming sounds were presented through headphones during a tactile detection task. Results showed that looming sounds reduced tactile RTs by 20–25 ms compared to static sounds, replicating a well-established multisensory effect linked to peripersonal space. These findings present a robust method for validating smartphones for cognitive research and demonstrate that high-precision audio–tactile paradigms can be reliably implemented on mobile devices. This work lays the groundwork for rigorous, scalable, and ecologically valid multisensory behavioral studies in naturalistic environments, expanding participant reach and enhancing the relevance of multisensory research.

## 1. Introduction

Smartphones have become ubiquitous in our daily routines. In 2024, the USA had the highest number of users (98%) of the population owning a smartphone [1] followed by the EU (96%) [2]. Recent studies used smartphones to enable researchers to collect behavioral data in participants’ everyday environments, thereby enhancing the ecological validity of such experiments [3,4]. This approach enables the widespread deployment of cognitive and psychophysical tests outside the laboratory while maintaining rigorous data quality, thus reaching more diverse and naturalistic participant samples.

More than a decade ago, Geoffrey Miller anticipated the transformative potential of smartphones in psychological research, highlighting their ubiquity and built-in sensors as key enablers of ecologically valid data collection [5]. Using this device—which most people already possess and that is equipped with a suite of high-performance sensors—to deploy experimental protocols remotely has the potential to greatly expand the scale of research on human behavior and emotion. In experimental psychology, online studies have already demonstrated the ability to reach a broader population. Platforms such as Gorilla Experiment Builder and LIONESS Lab enable researchers to deploy complex surveys on the web, significantly expanding participant reach beyond the lab [6,7] by leveraging computers and web technologies to enhance experiment scalability [8].

Cognitive experiments are traditionally conducted in laboratory settings, where participants respond to stimuli presented on computers or specialized devices. In research on human behavior and emotion, a widely used approaches involves measuring reaction times (RTs). Many such experiments rest on psychophysical methods and require high-precision timing for both stimulus presentation and response recording [9]. Research on multisensory integration presents another increasing complexity in precisely delivering stimuli, which typically requires the synchronized delivery of stimuli in several sensory modalities. The need for tight synchronization is well recognized [10,11]; if not carefully controlled, delays between sensory modalities delivery can occur—particularly on smartphones [12]. Prior work has documented such limitations. For example, Arthurs and colleagues found that uncorrected RTs recorded on an iPhone during a simple psychomotor vigilance task were overestimated, underscoring the need to assess hardware constraints in mobile-based experimentation [13].

Our team has conducted several laboratory-based experiments on audio–tactile integration, focusing on participants’ RTs to a tactile stimulus delivered to the hand while they simultaneously hear an auditory stimulus. The aim of this paradigm is to examine how the detection of a tactile stimulus is modulated by the concurrent presence of a looming sound [14,15]. Notably, when the tactile stimulus is presented with a longer delay relative to the onset of the auditory stimulus—corresponding to a sound perceived as approaching—the detection of touch is typically faster. This facilitation is interpreted as an indicator that the sound source has entered the individual’s peripersonal space. Peripersonal space refers to the dynamic region immediately surrounding the body, which plays a key role in multisensory integration and action preparation. Its boundaries are known to be highly plastic, adjusting according to emotional states, social context, and individual motor capabilities [14,16,17,18,19,20,21]. To date, most studies investigating this phenomenon have been conducted under controlled laboratory conditions, ensuring precise stimulus presentation and precise RT measurements. However, there is a growing interest in exploring peripersonal space in more ecological and diverse contexts, to better understand its role in everyday perception and behavior. Using smartphones could enable the investigation of peripersonal space outside the lab. However, it remains unclear whether these devices are suitable for obtaining robust and reliable results with the RT-based audio-tactile paradigm used to study peripersonal space.

In the present study, we focused specifically on Android smartphones, which account for the majority of the global market share (76%) [22], and are generally widely available and more affordable than alternative platforms. In addition, Android devices provide greater flexibility for low-level system access, making them particularly suitable for implementing and validating psychophysical paradigms that require fine-grained control over stimulus delivery and response logging. Rather than comparing a wide range of devices, our objective was to determine the feasability of reliably measuring RTs in an audio–tactile paradigm using a smartphone. The purpose of this development was to enable the deployment of such paradigms outside controlled laboratory settings, thereby collecting more ecological data while preserving experimental rigor.

The key contributions of the present study are the following:We propose a reproducible methodology to assess the suitability of Android smartphones for conducting audio–tactile RT-based paradigms. This methodology includes evaluating the device’s timing precision in synchronized auditory and tactile delivery, as well as in RT logging.We introduce a novel approach to improve RT measurement precision on smartphones by combining touchscreen and accelerometer data.An Android app–Dynaspace– was developed to implement audio–tactile interaction tasks, such as the one used to study of peripersonal space.Experimental results show that multisensory effects on RTs typically observed under controlled laboratory conditions can be replicated using a top-performing Android smartphone.

The remainder of this paper is organized as follows. The few previous research work on multisensory experimentation using smartphones are reviewed in Section 2. Then, we describe the software environment and measurement chain that we used in the present work (Section 3). In Section 4, we report the foundational stage of our work that consisted in developing a methodology to assess a smartphone hardware performance in delivering synchronized auditory and tactile stimuli and in collecting RT data. Section 5 describes the method we implemented to enhance smartphones’ RTs precision measurement by leveraging data from both the touchscreen and accelerometer sensors. In Section 6, we apply these methods to five Android models in order to evaluate their suitability for audio-tactile paradigms. In Section 7, we report a behavioral study conducted to test whether multisensory effects on RTs that are typically observed under controlled laboratory conditions could be replicated using a top-performing Android smartphone. We developed an app (Dynaspace) to program the audio-tactile paradigm measuring peripersonal space [14,15] and conducted a behavioral experiment on healthy participants. We discuss the implications of deploying such paradigms beyond the laboratory and outline perspectives for ecological and scalable research in cognitive and behavioral sciences in Section 8. Finally, we report the limitations of the present work as well as directions for future work (Section 9) and conclude (Section 10).

## 2. Related Work

Some work has already been conducted with the aim of taking multisensory psychophysical experimentation outside the laboratory thanks to the use of smartphones [23,24,25,26,27]. For example, PsySuite is an Android application enabling the delivery of multisensory stimuli with high temporal precision [23]. However, it was not designed for paradigms involving RTs and does not assess the timing precision of participant responses. Conversely, Marin-Campos and colleagues introduced StimuliApp, an open-source iPhone/iPad application that can present auditory and visual stimuli and collect responses with millisecond precision, illustrating that high-quality psychophysical testing is feasible on consumer smartphones. StimuliApp offers millisecond-level control of stimulus presentation on iOS [24]. Concerning the measure of RTs, the precision of log was dependent on the smartphone touchscreen refresh rate. Yet, a large-scale evaluation on device-related latencies by Nicosia and colleagues [28] revealed substantial variability in RT measurements across smartphone models and operating systems. Using Swift for development on iOS and Java on Android, the study found that latencies were highly different from one device to another and often exceeded the magnitude of typical experimental effects (commonly differences of 20–50 ms). These findings underscore the importance of accounting for hardware differences [29] when using smartphones for behavioral experiments measuring RTs. They also highlight the need for improved methods to collect RTs via touchscreen with a greater precision.

## 3. Development Software and Measurement Chain for the Proposed Methods

In this section, we first present the development environment used to create the app for assessing smartphone hardware performance, as well as to develop Dynaspace, the app designed for conducting audio–tactile paradigms in Section 3.1. Then, Section 3.2 describes the measurement chain employed to evaluate smartphone hardware performance and quantifies the timing error introduced by this measurement system itself.

### 3.1. Environment of Protocol Development

All applications used in this research work were developed in Kotlin, the native programming language for the Android platform. Native applications are generally preferred when performance and close interaction with device hardware are required [30]. Frameworks such as React Native or web-based implementations introduce additional abstraction layers, which can lead to increased latency and reduced timing precision. Prior benchmarking studies have found that purely web-based applications typically exhibit greater timing variability than native code [9], supporting our choice of a native Android implementation for optimal control over timing and precision.

This approach allowed us to interact closely with the operating system’s memory management and real-time processing capabilities [31]. Moreover, the development pipeline incorporated C++ for efficient thread management and audio processing, ensuring high performance and low-latency signal handling. To improve resource efficiency, we adopted an object-oriented programming (OOP) approach, instantiating only essential components at runtime. Kotlin served as the interface layer, managing class calls and object instantiations and providing structured access to core components.

Only a limited number of external libraries were added to manage the audio buffer, implement Open Sound Control (OSC), and provide precise timestamps. For low latency audio-handling, we used Oboe, identified as the most efficient solution by Inuggi et al. in PsySuite [23]. This Google-maintained library offers fine-grained control over the audio buffer and automatically selects the most appropriate audio driver available on the device. On smartphones running Android 8.1 (API level 27) or higher, Oboe defaults to AAudio; on earlier versions, it falls back to OpenSL ES. None of the five smartphones tested in this study required the fallback driver.

To enable seamless communication between the smartphone and a computer, we used the OSC (Open Sound Control) protocol, which enables data transmission across multiple devices over a local area network (LAN). In our setup, a computer running Max/MSP sent control commands to the smartphone, while the smartphone concurrently transmitted collected data back to the computer via OSC. This configuration centralized all measurements data on the computer, facilitating real-time monitoring and post-processing.

### 3.2. Evaluation of Timing Error Introduced by the Measurement Chain

All timing validation measurements were performed using a digital audio interface. Auditory stimuli, tactile vibrations, and user responses were recorded as audio signals via the smartphone’s headphone jack and a piezoelectric microphone, which captured both the vibratory stimulus and the participant’s response. To evaluate potential timing error introduced by the measurement system itself—comprising the computer, the Max/MSP environment, and the audio interface—we designed a dedicated evaluation setup depicted in Figure 1.

In this setup configuration, an initial sound is captured by the piezoelectric microphone (signal in red) and routed to the audio interface’s output. Upon detection in Max/MSP, a click sound was immediately triggered and sent back out through the interface’s output. The input (in yellow) and output (in blue) signals were then recorded and compared using an oscilloscope.

As illustrated in Figure 2, the time difference between the piezo input (yellow trace) and the output click (blue trace) showed a variability of approximately 3 ms. This round-trip latency accounts for both input and output buffering within the system. To estimate the unidirectional timing variability introduced by the measurement chain, we divided this total by two, yielding a standard deviation of 1.5 ms.

This value establishes a critical baseline: any delay variation smaller than 1.5 ms cannot be reliably attributed to the smartphone under test, as it falls within the noise floor of the measurement setup itself. Hence, 1.5 ms serves as the lower bound of temporal resolution when evaluating the timing accuracy and precision of smartphones in the following tests.

## 4. Assessment of a Smartphone Performance for Audio–Tactile Stimuli Synchronized Delivery

In this section, we assessed the device’s capacity to deliver synchronized auditory and tactile stimuli with high precision. Precise temporal coordination of auditory and tactile signals is essential for studying multisensory integration processes in humans. Even small latencies between modalities can influence perceptual and cognitive responses [12]. Therefore, it is critical to evaluate both the accuracy (the discrepancy between intended and actual onset) and the precision (the variability across repeated trials) of auditory and tactile stimulus delivery on smartphones.

To this end, we conducted a series of measurements involving the serial presentation of unisensory (auditory or tactile) and multisensory (audio–tactile) stimuli using a Google Pixel 2 XL smartphone. The auditory stimulus consisted of a Dirac impulse encoded in a waveform file, while the tactile stimulus was a 50 ms vibration generated by the smartphone’s vibromotor—this duration being the minimum required to reliably activate the motor. Auditory output was recorded through the smartphone’s jack port (connected to Input 1 of a digital audio interface), and the tactile stimulus was captured using a piezoelectric microphone attached to the smartphone and connected to Input 2 of the same audio interface, the connections are shown Figure 3a. All signals were recorded using the Audacity audio recording software (version 3.7.0) and subsequently analyzed with a custom Python script.

In the unisensory conditions (auditory or tactile), stimuli were presented every 500 ms (interstimulus interval, ISI) for 5 min, and the interval between successive stimuli was measured. In the multisensory conditions, pairs of auditory and tactile stimuli were delivered repeatedly, with the tactile stimulus presented 500 ms after the auditory one. The temporal interval between the two modalities was then measured to assess synchronization accuracy described in Figure 3b.

The simplified pseudo-code below describes the logic of the three protocols loops:while(start){        emitVibration();        delay(500);}


while(start){
        emitSound();        delay(500);
}



while(start){
        i++;        if(i%2)                emitSound();        else                emitVibration();        delay(500);
}


This setup enabled a precise assessment of the temporal properties of unisensory and multisensory stimulus delivery. On Table 1, the results regarding the measured timing difference between intended and actual onset times of stimuli (Δ) are reported. The Mean Δ corresponds to the averaged timing difference which reflects the accuracy of stimuli delivery. It is important to evaluate this mean difference because, if stable, it can be corrected by adjusting the timing of command execution accordingly. Thus, any high value does not necessarily make reliable psychophysical testing impossible. The SD Δ (standard deviation Δ) corresponds to the variability of the timing differences and reflects the precision of stimulus delivery. This is the variable that is crucial: high variability, i.e., low precision of stimulus delivery across trials preclude from getting robust and exploitable experimental results.

As summarized in Table 1, the difference between intended and actual onset times was minimal for tactile stimuli, indicating high accuracy. Auditory and audio–tactile conditions showed, however, slightly reduced accuracy. Nevertheless, decisively, the variability in stimulus onset times across trials was low in all sensory conditions, demonstrating high precision. These results indicate that any fixed onset offsets can be systematically corrected by adjusting the timing of command execution, thereby ensuring reliable stimulus delivery for psychophysical testing.

## 5. Enhancement of Reaction Time Measurement Precision

We aimed to evaluate whether a smartphone can measure a user’s response times with sufficient precision for RT sensitive psychophysical paradigms. In such tasks, participants are typically instructed to respond to stimuli as quickly as possible, and the timing of their response must be recorded with high temporal fidelity. However, even high-end smartphones are equipped with touchscreen sensors that generally operate at a sampling rate of approximately 120 Hz, introducing a temporal uncertainty of around 8.3 ms [28,32]. This level of imprecision can be limiting, particularly in audio–tactile paradigms used to assess peripersonal space, where differences on the order of 20 ms may be meaningful.

To enhance RT measurement, we developed a method that combines input from both the touchscreen and the accelerometer sensors. Touchscreen timestamps alone are limited by sensor refresh rates and are subject to delays introduced by the operating system’s processing pipeline. In contrast, the built-in accelerometer of the Pixel 2 XL (used in this study) offers a refresh rate of approximately 500 Hz, which corresponds to a temporal resolution of 2 ms. Accelerometers have been shown to be reliable in a range of applications, including fall detection in older adults and validation against motion capture systems, demonstrating their suitability for time-sensitive behavioral measurements [33,34].

Unlike touchscreen events, the accelerometer directly captures the physical impulse generated when the user taps the screen. This hardware-based detection bypasses system-level processing delays and provides a more accurate timestamp. To extract meaningful RTs from the accelerometer data, we developed a peak-detection algorithm inspired by methods commonly used in audio signal processing. The goal was to identify the sharp movement induced by a participant’s finger tap.

The accelerometer signal was first smoothed using an Exponentially-Weighted Moving Average (EWMA) filter, which reduces high-frequency noise while preserving temporal dynamics:(1)$$y_{n}=\left|x_{n-1}-x_{n}\right|\;\left(1-\alpha \right)+\alpha \;y_{n-1}$$

We then computed the magnitude of the acceleration vector:(2)$$magnitude=∥Acc_{xy}∥=\sqrt{Acc_{x}^{2}+Acc_{y}^{2}+Acc_{z}^{2}}$$

*y_n_* is the smoothed difference between successive samples, and corresponds to an exponentially weighted moving average of the absolute signal difference between consecutive samples, and alpha coefficient controls the weighting of past samples in the smoothing process. A threshold was applied to the magnitude signal to detect peaks corresponding to intentional screen taps. The threshold and smoothing parameters were calibrated to distinguish intentional taps from irrelevant micro-movements of the device.

To further minimize false detections and improve reliability, we combined data from both the touchscreen and the accelerometer. Once a touchscreen response is logged, we search within a temporal window of approximately 8.33 ms for the most recent accelerometer peak. This hybrid approach capitalizes on the strengths of both sensors: the direct detection of physical impact via the accelerometer, and the event-logging reliability of the touchscreen interface (see Figure 4). The complete implementation of this method, including code for data acquisition, filtering, and response detection, is available in our public repository (onsetDetector, https://forge-2.ircam.fr/roussel/dynaspace/-/blob/main/DynaSpaceIrcam/app/src/main/cpp/onsetDetector.cpp?ref_type=heads, accessed on 25 September 2025).

If the refresh rates of the touchscreen or accelerometer were higher than those of the device tested here, our detection method would remain valid, since the alignment procedure relies on precise timestamps and relative peak detection. In fact, higher sampling frequencies would only improve the temporal resolution of accelerometer peaks or touchscreen events, further increasing confidence in the RT estimates. Conversely, devices with substantially lower refresh rates could introduce unacceptable timing uncertainty; however, this risk can be mitigated by an acknowledgment of the hardware specifications and by excluding devices whose sensors do not meet the required temporal precision.

## 6. Assessment of Five Smartphones’ Performance Suitability for Audio-Tactile Reaction Time Paradigms

In this section, we first present the complete methodology proposed to assess the suitability of smartphones for audio–tactile RT paradigms in Section 6.1. Section 6.2 reports the results obtained by applying this methodology to five Android smartphones. Finally, Section 6.3 discusses these results in light of performance indicators available in the device documentation.

### 6.1. Methodology

We developed a custom protocol to evaluate whether smartphones are suitable for audio–tactile paradigms requiring precise RT measurements. Inspired by previous setups in Kaaresoja and Brewster [12] and Arrieux and Ivins [29], our installation placed the smartphone on a stable surface to minimize extraneous movement. A 3D-printed robotic finger made from PLA, controlled by an ESP32 microcontroller, was used to simulate a participant’s screen tap in response to stimuli (see Figure 5a).

A dedicated test application—latencePhoneApp—was developed in Kotlin to deliver audio–tactile stimuli and log RTs based on accelerometer-based peak detection. The application is publicly available via the Dynaspace repository (Dynaspace Git, https://forge-2.ircam.fr/roussel/dynaspace/, accessed on 25 september 2025).

Each trial began with the smartphone delivering an auditory stimulus, followed by a vibratory tactile stimulus after a randomized interstimulus interval (ISI) between 0.5 and 1.5 s. These stimuli were detected by Max/MSP using an audio interface: the auditory output via the headphone jack and the vibration via a piezoelectric sensor attached to the smartphone. Upon detecting both events, Max/MSP sent an OSC message to the ESP32, triggering the robotic finger to tap the smartphone screen illustrated in Figure 6.

This setup allowed us to emulate a fully self-contained, out-of-lab scenario using only the smartphone’s built-in hardware to both generate stimuli and collect response times. In parallel with internal smartphone logging, all events were also recorded as analog audio signals through the digital audio interface, providing a time-resolved external reference against which smartphone timestamps could be compared.

The auditory stimulus consisted of a brief Dirac impulse (WAV format) played through the smartphone’s headphone jack. The tactile stimulus was a 50 ms vibration delivered via the built-in haptic motor, captured by a piezoelectric microphone placed at the back of the device—where the vibratory signal was most prominent. To ensure tight synchrony, the application pre-loaded the audio sample and initialized the timer within the buffer-loading routine.

The simplified pseudo-code below outlines the application logic:while(start){        delta_1 = rand(500,2500);        sendOSC(delta_1);        emitSound();** **        delay(delta_1);        emitVibration();** **        while(!catchOnset.get()){}        timeOnset = catchOnset.get();        catchOnset.reset();** **        delta_2 = timeOnse - delta_1;        sendOSC(delta_2);}

Internally, the smartphone recorded three key timestamps:

t0: Onset of the auditory stimulus (triggered from the audio buffer)

t1: Onset of the vibratory tactile stimulus

t2: Moment of screen tap by the robotic finger (detected via accelerometer)

From these, the intervals t0−t1 (stimulus onset delay) and t1−t2 (RT) were computed. In parallel, the same events were recorded externally using the audio interface, producing corresponding time intervals t0’−t1’ and t1’−t2’ based on waveform analysis. All internal timestamps were transmitted to Max/MSP via OSC, which acted as the centralized logging system.

To introduce variability, the auditory–tactile delay (t0−t1) was randomized between 0.5 and 1.5 s. Once Max/MSP received the vibration onset, it introduced an additional random delay between 1 and 2 s before triggering the robotic tap. This sequence was repeated for more than 1000 trials.

All data were centralized in Max/MSP and exported as a CSV file containing both the smartphone-based (t0−t1, t1−t2) and audio-based (t0’−t1’, t1’−t2’) measurements. We computed the differences between corresponding pairs, i.e., (to−t1−t0’−t1’) and (t1−t2−t1’−t2’), to evaluate the accuracy of the smartphone’s timing. To assess precision, we calculated the standard deviation of each set of differences across trials.

These measurements were repeated across multiple smartphone models, each tested in over 1000 automated trials to ensure robust and statistically meaningful comparisons.

### 6.2. Results

With the aim to identify at least one smartphone suitable for use in audio–tactile paradigm, we applied the method described in Section 6.1 to assess the performance of five different smartphones (see Table 2 for a description of the tested models). The objective was to evaluate their suitability for use in audio–tactile paradigms, specifically with respect to the temporal accuracy and precision of stimulus delivery and RT collection. We focused specifically on Android smartphones, which are widely available and generally more affordable than alternative platforms. In addition, Android devices provide greater flexibility for low-level system access, making them particularly suitable for implementing and validating psychophysical paradigms that require fine-grained control over stimulus delivery and response logging.

The results indicate that certain consumer-grade smartphones are capable of delivering multisensory stimuli and capturing RTs with sufficient temporal resolution for use in experiments on audio–tactile integration.

As can be seen in Figure 7 and Figure 8, the evaluation focused on two key synchronization criteria: (1) the precision of delay between auditory and tactile stimuli and (2) the precision of tactile RT. The Motorola device demonstrated high precision in the tactile RT measurement but showed substantial variability in the synchronization between auditory and tactile stimuli. In contrast, smartphones such as the Pixel 2 XL and the Fairphone 3 (bottom left quadrant of Figure 7) exhibited very good performance across both criteria. Their timing precision ranged from 5 to 10 ms in both modalities, making them viable candidates for deploying audio–tactile paradigms that require precise stimulus timing and response logging.

### 6.3. Indicator of Smartphone Performances

The “Audio latency” section of the Android NDK documentation [31] indicates that audio performance is highly variable across Android devices. However, it also provides several useful indicators that can help identify potentially suitable smartphones for low-latency experimental use. These include the following:

hasLowLatencyFeature: a boolean flag indicating whether the device supports low-latency audio output.

hasProFeature: a boolean flag indicating support for professional-grade audio features.

HighestDirectReportRateLevel: an indicator of the accelerometer’s maximum sampling frequency.

The latter variable is only available for devices running API level 31 (March 2022) or above and may return RATE_STOP on unsupported models, suggesting that the sampling rate cannot be directly queried. In such cases, accelerometer data must be collected in real-time, and the sampling rate inferred from the frequency of received data.

To further evaluate smartphone performance, we also measured the audio input/output buffer latency using a method similar to the one employed for assessing the measurement error introduced by the digital audio interface. A Dirac impulse was played through the smartphone speaker, triggering a timer. The impulse was simultaneously captured by the device’s microphone, which then stopped the timer. This round-trip latency measurement allowed us to estimate the combined input/output buffer delay, with the one-way buffer latency approximated as half of the total round-trip time.

The smartphones demonstrating the highest temporal precision in Figure 7 and Figure 8 were also those for which both hasLowLatencyFeature and hasProFeature returned true (see Table 3). These devices exhibited the lowest input/output buffer latencies and supported accelerometer refresh rates exceeding 400 Hz. Notably, all high-performing smartphones shared the same processor architecture—a Qualcomm Snapdragon CPU (see Table 2).

These findings suggest that developer-accessible indicators, such as system features and sensor capabilities, can serve as reliable proxies for estimating a smartphone’s suitability for conducting audio–tactile RT paradigms. Based on these indicators, it is possible to exclude devices likely to exhibit suboptimal performance without executing the full validation protocol. One of the main advantages of this approach is its accessibility: all assessments were carried out using only the smartphone’s embedded hardware and sensors, without requiring any specialized laboratory equipment.

## 7. Behavioral Validation

To verify that RT data collected with a smartphone in an audio–tactile task [14,15] could be comparable to data obtained with standard PC laboratory setups, we set up a version of the task in an Android app (Dynaspace). We chose the smartphone that had the best measured performances during hardware assessment, i.e., the Google Pixel 2 XL model (see Section 6.3), and used it to replicate an audio–tactile experiment [14] on a sample of 20 healthy participants.

The methods for this audio–tactile behavioral study are described in Section 7.1, and the results are presented and discussed in Section 7.2.

### 7.1. Methods

In this section, we first describe the app developed to conduct the audio–tactile task on Android smartphones (Section 7.1.1). We then present the sample of participants recruited to perform the audio–tactile task using the app (Section 7.1.2). The experimental setup, stimuli, and procedure are detailed in Section 7.1.3 and Section 7.1.4.

#### 7.1.1. Dynaspace: Implementation of the Audio–Tactile Paradigm

We developed the Dynaspace app to conduct the audio–tactile task measuring peripersonal space on an Android smartphone. The application’s home screen displays the task instructions, followed by a start button that initiates a 5-min experimental block. The block begins with a brief practice phase consisting of five training trials. Participants’ responses during this phase are not recorded, as its sole purpose is to familiarize them with the task. After the training phase, 72 experimental trials are presented, and responses are recorded (see Section 7.1.4). The trials were designed to replicate the audio–tactile stimulation protocol used in the study by Hobeika and colleagues [14]. Each trial begins with the onset of a sound file and then a tactile vibratory stimulus is delivered at different delays from sound onset. When participants respond to the tactile vibratory stimulus by tapping the screen, they receive visual feedback confirming that their response was registered—indicated by a change in screen color. At the end of the block, a prompt appears allowing the user to enter a filename for the output data. The results are saved as a CSV file on the device. Each row in the CSV corresponds to a single trial and includes a flag indicating the auditory stimulus type (“looming” or “fixed”), the timing of the tactile stimulus delivery, and the participant’s RT. Once saved, the file can be retrieved by connecting the smartphone to a computer; the CSV is stored in the smartphone’s accessible file directory. Before beginning the experiment with Dynaspace, we conducted pilot testing to refine the user interface and user experience (UI/UX), and to calibrate the accelerometer sensitivity for response detection (see Section 5).

#### 7.1.2. Participants

Twenty healthy participants (11 women and 9 men; age: m = 24.4 years, sd = 2.5; range [21:31]) were included in the study. Sample size was decided a priori based on previous work examining peripersonal space boundaries with the same audio–tactile paradigm [14,15,35,36]. All participants reported normal hearing and touch. None of them had a history of psychiatric disorders, neurological disorders, or were currently undergoing medical treatment. All of them were right-handed. All participants provided informed consent and were paid for their participation in the study, which was approved by the French Ethic Committee (CPP Nord-Ouest I-CPP n°2019-A01131-56).

#### 7.1.3. Experimental Setup and Stimuli

Participants were seated on a chair and equipped with headphones. They held the smartphone in their left hand, which was positioned in front of them as illustrated in Figure 9. The smartphone was the same model for every participant (Google Pixel 2XL) to avoid noise due to performance variability of the devices [37,38]. Auditory stimuli were presented through Beyer Dynamics DT770 headphones. The auditory stimuli were sounds of 3250 ms duration composed of a bursts train (44,100 Hz digitization; see [14] for a description of the bursts train). In order to simulate spatialized auditory sources, the bursts train auditory stimulus was processed in a Max/MSP (6.1.10) environment using the Spat library [39]. Using the same procedure as in previous work, the direct sound, early reflections, and a late reverberation were simulated [14,16,36]. The spatialization of the direct sound and of the early reflections was binaurally rendered using non-individual head-related transfer functions (HRTF) of the LISTEN HRTF database [40]. This procedure allows for manipulating the virtual auditory stimulus source location by rendering accurate auditory cues such as frequency spectrum, intensity, and inter-aural differences. The whole procedure was applied to the bursts train auditory stimulus in order to render two different auditory stimuli: a fixed auditory stimulus and a looming auditory stimulus. For both types of auditory stimuli, we simulated a sound source located in the frontal hemifield in the left hemispace: the direct auditory stimulus was spatialized with azimuth 45°. All participants confirmed that they could clearly locate auditory stimulus sources in the left hemispace. For the “fixed” auditory stimulus, the sound source remained static and located at 640 cm from participants’ head center. For the “looming” sound stimuli, the sound source was dynamic and approached participants’ head center from 640 cm to 20 cm, at ear-level and at a constant speed (210 cm.s^−1^). Tactile stimuli were delivered via the smartphone’s haptic motor, while participants held the device in their left hand. The tactile stimulus consisted of a vibration at 155 Hz for 50 ms, the minimum duration to activate the haptic motor on Android smartphones, with maximum amplitudes.

#### 7.1.4. Experimental Procedure

During the experimental session, participants were asked to tap on the screen of the smartphone each time they detected a tactile stimulus. They were asked to ignore the auditory stimulus and to respond as quickly as possible to the tactile stimulus while avoiding anticipation. We measured participant’s tactile RTs. For each trial, the auditory stimulus (fixed or looming) was presented for 3250 ms. The auditory stimulus was preceded by a 300 ms period of silence and followed by a randomly distributed period of silence between 1000 and 1400 ms. Two types of trials were defined: experimental trials, which included both auditory and tactile stimulation; and catch trials, in which the auditory stimulus was delivered but no tactile stimulus was presented. The catch trials were included to decrease participants’ expectation of the tactile stimulus in trials [41]. For experimental trials, the tactile stimulus was presented during the duration of the auditory stimulus. In the experimental trials, the tactile stimuli were delivered at different times from the onset of the sound. Temporal delays for the tactile stimulus were set as follows: (T1: 105 ms, T2: 1625 ms, T3: 2385 ms, T4: 2765 ms, T5: 2955 ms, T6: 3050 ms; see Figure 9a). When the auditory stimulus was a looming sound, the distance of the sound source when the tactile stimulus occurred depended on the delay between the onset of the looming sound and the delivery of the tactile stimulation—longer distances from the participant for shorter delays and shorter distances for longer delays. The spatial distributions of the sound source at each delay followed a logarithmic scale (640 cm, 320 cm, 160 cm, 80 cm, 40 cm, 20 cm), following the recommendations of Hobeika and colleagues [14]. In the conditions where the sound source remained fixed and distant from participants’ body, the delays between sound onset and tactile delivery were the same as in the looming sound condition, but the sound source itself remained at a far distance. This “fixed sound” condition allows for examining the effect of the expectancy of the tactile stimulus along the trial on RTs [14,41] and thus helps disentangle the spatial from the temporal effect of the sound stimulus on tactile processing. Participants were asked to perform six experimental blocks of 3 min duration. Between each block, participants took a short break, and a mandatory 10 min break was imposed halfway through the experimental blocks. Each block was composed of 48 experimental trials, and 24 catch trials. In total, participants performed 432 trials, 288 experimental trials, and 184 catch trials (33.3% of catch trials). They performed 24 repetitions for each of the 12 experimental conditions: 2 SOUND MOVEMENT (fixed/looming) and 6 DELAY (T1, T2, T3, T4, T5, T6).

### 7.2. Results

One participant was excluded from the analyses due to a high rate of misses (9.0% of miss, m ± sd of the sample: 1.4 ± 2.0% of miss). One additional participant had data missing for a block due to a procedural mishap by the experimenter. For the following analyses their data were excluded. The analyses were conducted on the 18 remaining participants (10 women and 8 men; age: m = 24.0 years, sd = 2.2; range [21:31]).

The processing of the data collected during the experiment was similar to the one conducted in previous studies [14,16,36]. Similar to the preceding experiments, participants demonstrated a high level of accuracy in performing the tactile detection task, as indicated by the low rate of misses (0.85 ± 0.76%). Consequently, the analysis focused on response time (RTs). RTs shorter than 100ms and 1000ms were removed from the analyses (0.37% of the trials), as we considered these RTs to be either too short or too long to accurately reflect participants’ performance in the speeded tactile detection task. The distribution of participants’ RTs exhibited skewness, a common feature in RT data [42]. To address this, we applied a natural logarithm transformation to each RT data before proceeding with subsequent statistical analyses on mean RTs [43]. Our objective was to determine the distance from which the sound source started to interact with the processing of the tactile stimulus, enhancing participants’ RTs. As previously mentioned, tactile expectation effects can also lead to a reduction in RTs, potentially complicating the interpretation of audio–tactile interaction effects in looming sound conditions. Therefore, our investigation aimed to identify the delay at which tactile RTs were shorter in the looming sound condition compared to the fixed sound condition. To analyze the impact of audio–tactile integration on tactile RTs, we compared the mean RTs between the looming and fixed sound conditions. We averaged ln(RT) in the fixed and looming sound conditions, separately for each delay condition (T1, T2, T3, T4, T5, T6). We then transformed back the mean ln(RT)s and conducted an ANOVA on the obtained values with the within-subject factors SOUND MOVEMENT (2 levels: fixed/looming) and DELAY (6 levels: T1, T2, T3, T4, T5, T6). The analysis revealed significant main effects of DELAY (F(5,85) = 105.6, *p* < 0.001, $\eta_{p^{2}}$ = 0.861) and SOUND MOVEMENT (F(1,17) = 23.3, *p* < 0.001, $\eta_{p^{2}}$ = 0.578) as well as a significant effect of the two-way interaction DELAY*SOUND MOVEMENT (F(5,85) = 4.7, *p* < 0.001, $\eta_{p^{2}}$ = 0.217). As illustrated in Figure 10, RTs for tactile stimuli at T1 and T2 were not significantly different in the fixed and looming sound conditions (*p* > 0.061 in both cases), while RTs at T3, T4, T5, and T6 were significantly shorter in the looming sound condition as compared to the fixed sound condition (*p* < 0.001 in all cases).

The data exhibited a typical pattern of audio–tactile paradigm data with a decrease in RTs as delay of tactile stimulation increased along the trial [14,15,16,17,18,20,35,36]. And crucially, the decrease in RTs was even larger when the sound was looming and approaching closer to the body. Similar to the study that we replicated [14], we found that RTs in the looming sound condition were shorter than those in the fixed sound condition at the four longest delays (T3, T4, T5, and T6), i.e., when the sound source was at the four closest positions to the participants. The RT acceleration was also of a similar magnitude, with a decrease of approximately 20–25 ms.

## 8. Discussion

In this study, we tested the feasibility of rigorously conducting an RT-based auditory–tactile paradigm using an Android smartphone. Although several studies have already deployed smartphone-based applications for multisensory research [23,24,25,26,27], most do not provide detailed descriptions of their multisensory stimulation, nor do they report validating the temporal precision of their setups. One notable exception is PsySuite, which has been evaluated for its ability to deliver multisensory stimuli; however, the issue of precise RT measurement was not addressed [23]. With the present work, we contribute to further advance the field of rigorous smartphone-based multisensory research by proposing a methodology to evaluate the suitability of Android smartphones for both the delivery of auditory–tactile stimuli and the precise measurement of RT. Additionally, we describe a method for enhancing the precision of RT measurement on mobile devices.

By testing five different smartphone models, we quantified each device’s ability to deliver synchronized auditory and tactile stimuli and to log response times (RTs) with high temporal precision. For all five devices, measured timing precision closely aligned with manufacturer specifications, indicating that basic hardware indicators can reliably guide initial device selection for experiments. This validation also confirmed that auditory and vibratory tactile stimuli could be triggered with a temporal onset difference of only a few milliseconds. For two smartphones in particular, the standard deviation of the synchronization error remained below 10 ms, an order of magnitude smaller than the typical human threshold for detecting multisensory asynchrony (approximately 50–100 ms), as reported in previous psychophysical studies [10,11].

Another important methodological advancement concerns the measurement of response timing. While smartphones have already proven capable of capturing behavioral and social activity through embedded sensors [38], our approach augments standard touchscreen event detection with accelerometer data. Because the accelerometer often samples at a higher rate than the touchscreen (typically around 100 Hz vs. 120–200 Hz or higher), this allows for finer temporal resolution. By combining the touchscreen’s input event with a high-resolution peak from the accelerometer signal, we were able to evaluate the exact moment of the participant’s response with ~4 ms of precision with the Google Pixel 2XL smartphone model. This strategy compensates for the touchscreen’s limited temporal granularity (which generally operates at 8–10 ms intervals) [44], enabling more accurate behavioral measurements. Furthermore, using the accelerometer to capture the onset of the vibratory tactile stimulus offers a potential internal validation method for confirming stimulus delivery. This could enable real-time quality control and automatic latency correction, especially useful in unsupervised or field-based studies. Integrating this feature into the application is a next step that would further improve both its robustness and autonomy.

Overall, our validated system demonstrates that smartphones are capable of delivering psychophysical experiments with high temporal precision and reliability. The Dynaspace app serves as a lightweight, low-cost alternative to traditional laboratory setups, reducing logistical barriers and increasing accessibility for both participants and researchers (Dynaspace Git, https://forge-2.ircam.fr/roussel/dynaspace/, accessed on 25 september 2025). In doing so, it enhances the external validity and ecological relevance of cognitive data collection. This enables the deployment of cognitive protocols, including RT-based and multisensory integration paradigms, in ecologically valid contexts outside the lab. For instance, the app can be used to test participants with limited mobility or those located in geographically remote regions, using only a device that most already own. It also enables the simultaneous testing of multiple participants across diverse settings, including schools, clinics, and public spaces. As demonstrated by Bignardi and colleagues, cognitive assessments conducted on tablets in classroom environments yield results comparable in reliability and validity to one-on-one lab testing [45].

Finally, this mobile approach offers promising applications for behavior analysis by capturing RT. RT is not only a measure of sensorimotor coordination—it is also sensitive to emotional and affective states such as fear, anxiety, or perceived threat [16,17]. The system thus enables the study of peripersonal space and its modulation by affective factors [46] in real-world, dynamic contexts. RT-based paradigms can be implemented in crowded or emotionally salient environments, providing insights into embodied emotional cognition. By extending psychophysical testing into naturalistic settings, this methodology has the potential to contribute to mobile mental health assessments and to advance the ecological validity of research on emotional reactivity and multisensory processing.

## 9. Limitations and Future Work

This work has several limitations, primarily due to its role as a necessary first step in exploring the potential of smartphone devices for conducting psychophysical, RT-based multisensory research outside the lab. First, since we aimed to identify at least one smartphone with performance levels suitable for precisely presenting multisensory stimuli and recording participants’ RTs, we limited our testing to five low-cost Android models with three different operating system versions. Among the devices, we found some that met the necessary performance requirements. This suggests that such smartphones could be distributed to participants for use in more ecologically valid experimental settings. A logical next step is to leverage the fact that most participants already own a smartphone. To fully take advantage of this for experimental purposes, future research should aim to characterize a broader range of smartphone models and operating system versions. The methodology we propose here can be directly applied to assess other Android smartphones and could be adapted for iOS devices by using the same setup and developing a dedicated Apple app.

Second, we did not explicitly address potential sources of variability related to device state, such as low battery levels or interference from background applications. While this was beyond the scope of the present work, future studies should investigate these factors to further enhance experimental control when using smartphones.

Third, to assess whether a smartphone could reliably capture a typical behavioral multisensory phenomenon usually measured in lab-based setups, we replicated a previous study [14]. The participant sample, experimental environment, and participant posture (seated in a room) were deliberately matched to those used in the original study. Now that we have demonstrated that the multisensory effect can be replicated using a smartphone, the next step should expand testing to a larger, more diverse sample of participants that better represents the general population. Crucially, this next phase will also involve deploying the audiotactile paradigm in more ecological environments and real-world situations—ultimately aligning with the broader goal of bringing multisensory research outside the lab.

## 10. Conclusions

In this work, we propose a reproducible protocol to assess smartphone precision for deploying audio–tactile paradigms, encompassing both stimulus delivery and RT measurement. In addition, we identified an Android-accessible variable that can serve as a built-in indicator of device performance. By combining touchscreen and accelerometer data, we introduce a novel method that reduces the inherent limitations of RT measurements on smartphones due to its touchscreen refresh rate. Our approach achieves performance close to laboratory-grade setups and demonstrates that a peripersonal space paradigm can be reliably reproduced outside the laboratory. Taken together, these advances provide a practical foundation for deploying multisensory psychophysical experiments in more ecological and scalable contexts.

While the results of our study are very promising, they come with some limitations: the methodology has so far only been applied to Android smartphones and tested with a sample of young, healthy participants in a laboratory environment. Future work should aim to broaden this scope by adapting the methodology to iOS devices and exploring the use of the Dynaspace app with diverse population of participants, in more ecological and real-world contexts, where its potential for scalable multisensory research can be further assessed. 

## Figures and Tables

**Figure 1 sensors-25-06072-f001:**
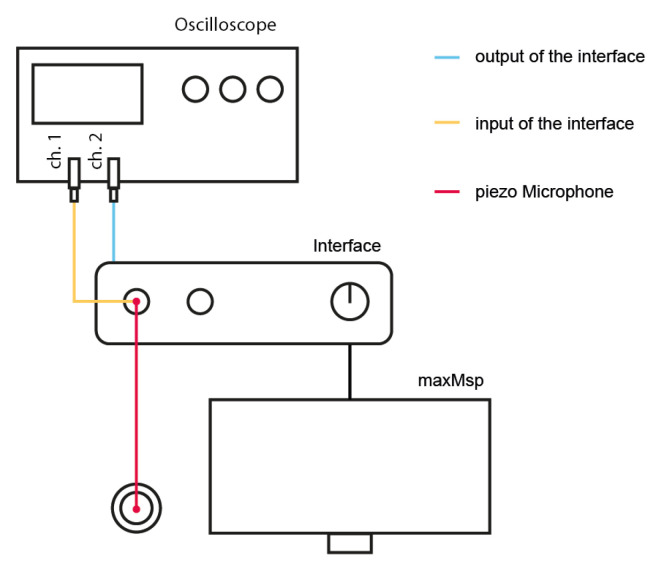
Experimental setup used to evaluate the timing error introduced by the MADIface XT audio interface and the Max/MSP environment. An impact was captured using a piezoelectric microphone. The signal was simultaneously routed to the input of the audio interface (red trace) and to Channel 1 (ch.1) of the oscilloscope (yellow trace). Upon detection in Max/MSP, an audio impulse was generated and sent through the output of the interface, which was connected to Channel 2 (ch. 2) of the oscilloscope (blue trace). The oscilloscope measured the time interval between the input and output signals, enabling the calculation of round-trip latency and the estimation of temporal variability introduced by the system.

**Figure 2 sensors-25-06072-f002:**
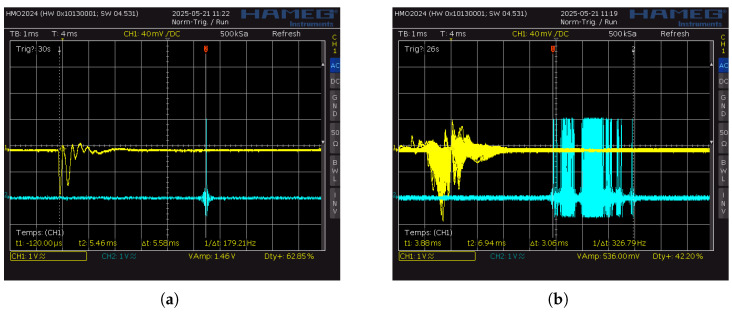
(**a**) Oscilloscope visualization of a single round trip latency measurement. The yellow trace (Channel 1) shows the incoming piezoelectric impulse; the blue trace (Channel 2) represents the corresponding audio output generated by the interface. (**b**) Multiple recordings were superimposed to evaluate the consistency of the round-trip timing. This accumulation of traces enabled the estimation of the system’s temporal precision, yielding an average variability of approximately 3 ms.

**Figure 3 sensors-25-06072-f003:**
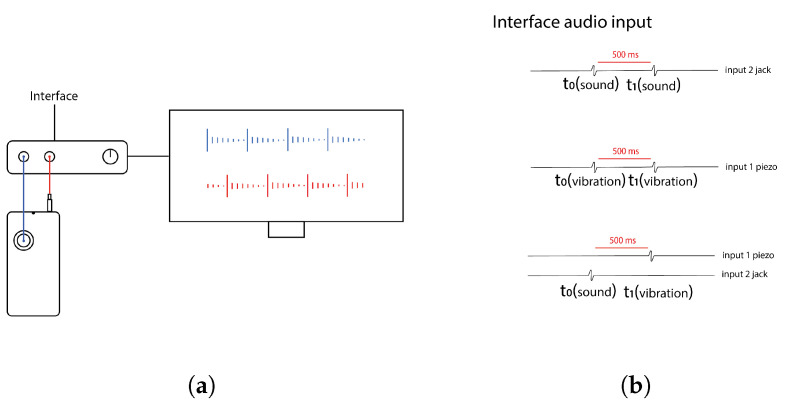
Assessment of accuracy and precision of stimuli delivery (**a**) The setup consisted of a smartphone, an audio interface, and a computer. The smartphone’s auditory output was recorded via a jack connection (red), while a piezoelectric microphone (blue) captured the tactile vibrations. Both signals were routed through the audio interface and recorded on the computer for offline analysis. This configuration enabled simultaneous monitoring of auditory and tactile outputs, allowing for precise assessment of the timing of the stimulus delivery. (**b**) Three protocols were used to evaluate the temporal precision of stimulus delivery. In the first, auditory stimuli were presented at fixed 500 ms intervals. In the second, vibratory tactile stimuli were delivered at the same intervals. The third protocol alternated between auditory and tactile stimuli, each delivered at 500 ms intervals. These protocols were designed to assess the temporal precision of the system across different sensory modalities and timing configurations.

**Figure 4 sensors-25-06072-f004:**
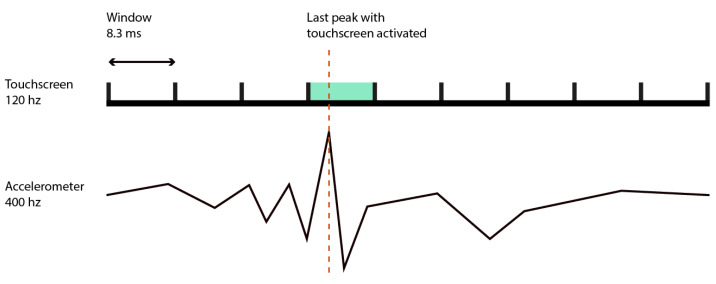
The figure shows two aligned timelines corresponding to the data streams from the touchscreen sensor (**top**) and of the accelerometer sensor (**bottom**), each sampled at different refresh rates. The red marker indicates the participant’s tap, while the green area represents the 8.3 ms temporal window during which the touchscreen logs the response. In the lower panel, a high-resolution peak in the accelerometer signal provides a more precise temporal marker of the physical interaction. By aligning the touchscreen event with the nearest accelerometer peak within this window, a more accurate estimate of the user’s RT is obtained.

**Figure 5 sensors-25-06072-f005:**
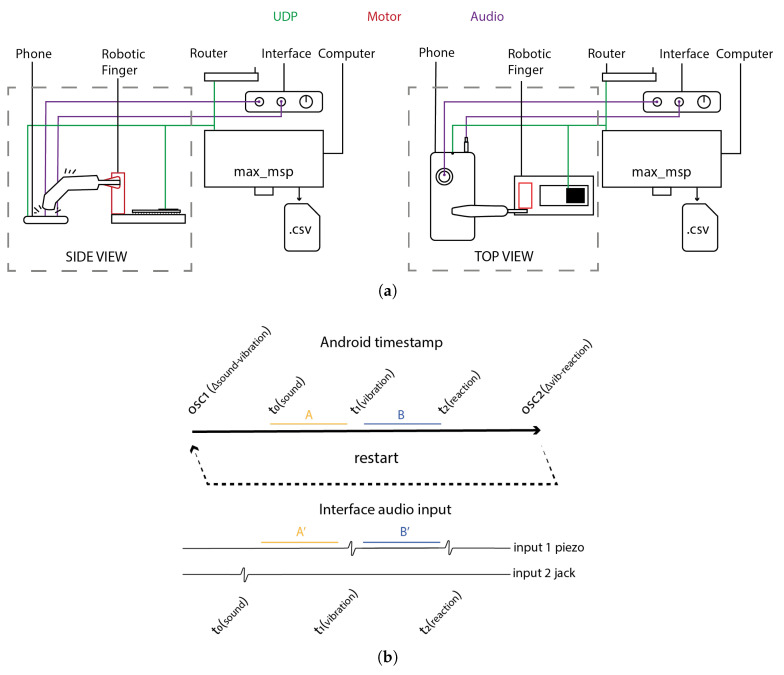
(**a**) Side and top view of the experimental setup used to evaluate smartphone timing performance in audio–tactile paradigms. A robotic finger, controlled by max_msp, was used to simulate a tactile response. Auditory stimuli (brief Dirac impulses) were recorded through the headphone jack using a digital audio interface. Vibratory tactile stimuli and robotic finger responses were captured via a piezoelectric sensor placed on the smartphone. (**b**) The smartphone transmitted its internal timestamps—corresponding to auditory stimulus onset, vibration onset, and response detection—via the OSC protocol. These timestamps were synchronized with external analog recordings to allow for comparison. The resulting paired intervals (e.g., A–A’ for stimulus delay and B–B’ for RT) were stored in a CSV file and used to evaluate the accuracy and precision of the smartphone’s internal timing mechanisms.

**Figure 6 sensors-25-06072-f006:**
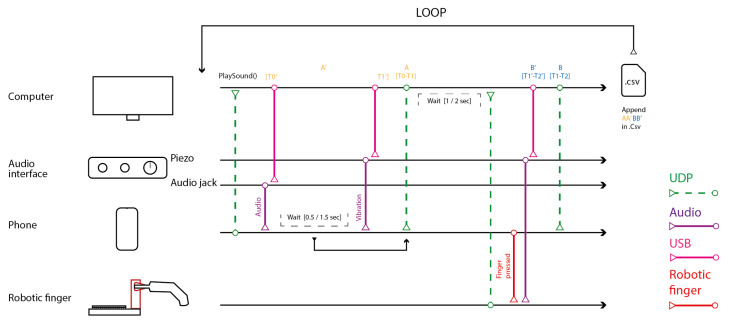
Schematic timeline of audio–tactile stimulus delivery and response logging. This timeline illustrates the sequence of operations used to assess the smartphone’s temporal precision in generating stimuli and recording responses. The experimental setup included four primary components: a digital audio interface, a computer running Max/MSP, a smartphone, and a robotic finger. Stimulus presentation and response execution followed a strictly ordered sequence to ensure temporal consistency. The smartphone’s internal timestamps were compared with external audio-based measurements recorded via the audio interface. This dual-logging approach enabled precise assessment of timing discrepancies and latencies between internally registered and externally observed events.

**Figure 7 sensors-25-06072-f007:**
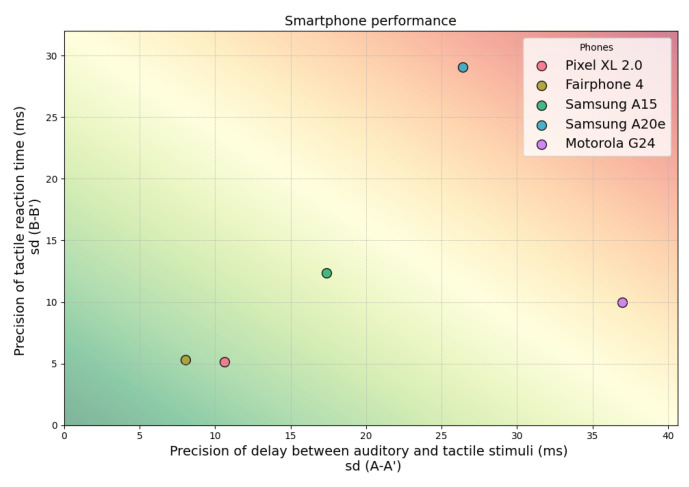
Smartphones’ measured performance. This figure depicts smartphones precision in the delivery of multisensory auditory and tactile stimuli as a function of the smartphone precision in collecting RTs. The closer the smartphone is positioned to the graph bottom-left corner, the better its performance for audio–tactile RT paradigms.

**Figure 8 sensors-25-06072-f008:**
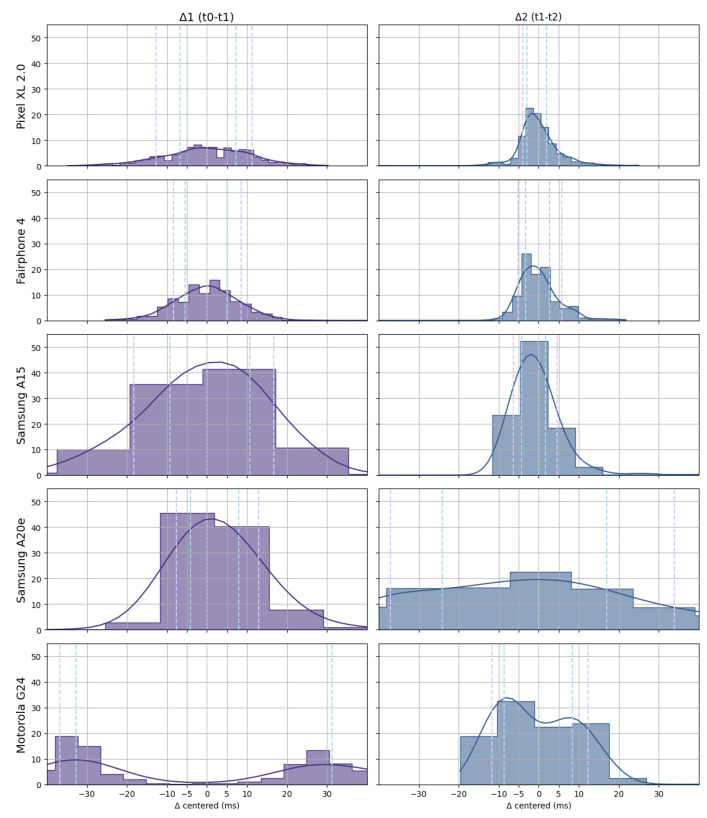
The distributions of measured references made with the digital audio interface and by each smartphone of $A\;A^{\prime }$(time between auditory and vibratory stimulus) and $B\;B^{\prime }$(time between vibratory stimulus and tactile response). Timing intervals are presented for each smartphone. For both measures, the 25th to 75th and 12.5th to 87.5th percentile ranges are shown individually per device, providing an overview of the precisions across multisensory deployment and precision on the measurement of RTs timing.

**Figure 9 sensors-25-06072-f009:**
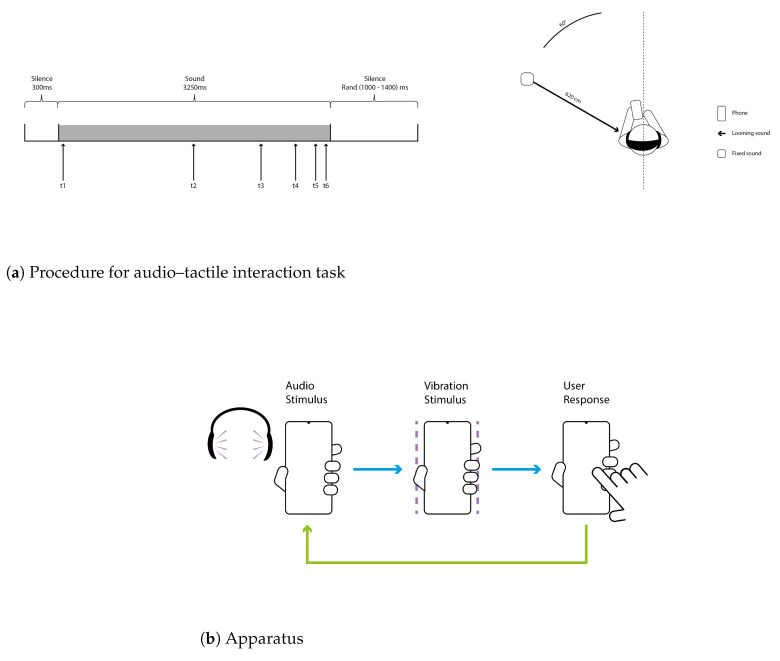
(**a**) Audio–tactile interaction task—(**Left** panel) Description of a trial. A trial begins with a period of silence followed by a 3250 ms-long auditory stimulus and ends with a second period of silence. (**Right** panel) Experimental setup. Participants received a tactile stimulation on their left hand (delivered using the haptic motor vibration of the smartphone) while a task-irrelevant spatialized auditory stimulus was presented to them in their left hemispace. The auditory stimulus could be static and located at a fixed position at 640 cm from the center of participant’s head (fixed sound) or looming towards participants from 640 cm to 20 cm distance (looming sound). The tactile stimulus was delivered at different delays from auditory stimulus onset (T1, T2, T3, T4, T5, T6). Hence, the looming auditory stimulus source was perceived at different locations with respect to the body when tactile stimulation occurred, i.e., far from the body at small temporal delays and close to the body at long temporal delays. (**b**) The figure represents the continuous trial loop used in the audio–tactile interaction task. Each trial follows a fixed sequence: an auditory stimulus is played, while it is playing, a vibratory tactile stimulus is delivered after a variable delay, the participant responds by tapping on the smartphone screen. At the end of the trial, the loop resets and the next auditory stimulus begins. This cycle repeats automatically until all 76 trials are completed.

**Figure 10 sensors-25-06072-f010:**
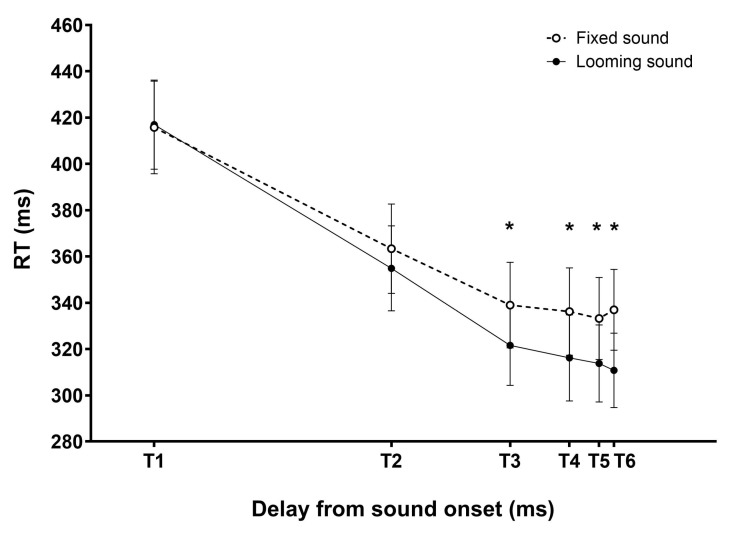
Audio–tactile interaction experiment results—Analysis of mean tactile RT. This figure depicts participants’ (n = 18) mean tactile RTs (RTs ± SEM) in the looming (filled black circle) and fixed (open circle) sound conditions, as a function of the delay between the tactile stimulation and the sound onset. Audio–tactile integration is evidenced by shorter RTs in the looming as compared to fixed sound condition; here, at T3, T4, T5, and T6 delay conditions (as indicated by asterisks).

**Table 1 sensors-25-06072-t001:** Accuracy and precision of auditory and tactile stimuli delivery.

	Accuracy			Precision
	Mean Δ (ms)	Min Δ (ms)	Max Δ (ms)	SD Δ (ms)
**Auditory**	+12.66	−0.021	+2	0.963
**Tactile**	+1.41	−19.833	+23.125	3.625
**Auditory + Tactile**	+32.085	+28.776	+37.188	0.2001

**Table 2 sensors-25-06072-t002:** Smartphones models characterization.

Manufacturer	Model	Year	Android	RAM (GB)	Processor
**Google**	Pixel 2 XL	2017	v11	4	Snapdragon 835
**Samsung**	Galaxy A20e	2019	v11	3	Exynos 7884
**Samsung**	Galaxy A15	2023	v14	4	MediaTek Dimensity 6100
**Fairphone**	4	2021	v13	8	Snapdragon 750G
**Motorola**	G24	2024	v14	4	MediaTek Helio G85

**Table 3 sensors-25-06072-t003:** Comparison of specific smartphone-accessible variables that provide insight into devices performance.

	Samsung A20	Samsung A15	Fairphone 4 5G	Motorola G24	Pixel 2 XL
**in/out latency**	~120 ms	~80 ms	60 ms	~110 ms	30 ms
**hasLowLatencyFeature**	false	true	true	true	true
**hasProFeature**	false	false	true	false	true
**Fq accelerometer**	~100 Hz	320 Hz	>400 Hz	>400 Hz	>400 Hz
**Fq touchScreen**	120 Hz	90 Hz	60 Hz	90 Hz	120 Hz

~ is used to notify this analysis is stuttering on some smartphones and always correlate with non-audio pro features.

## Data Availability

The conditions of our ethics approval do not permit public archiving of anonymised study data. Readers seeking access to the data should contact the corresponding author IVD.

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
