# Peer review of "Using Android Smartphones to Collect Precise Measures of Reaction Times to Multisensory Stimuli"

_sensors, 2025, doi:10.3390/s25196072_

Round 1

Reviewer 1 Report

Comments and Suggestions for Authors

The article addresses the use of smartphones to conduct behavioral research outside traditional laboratory environments, especially psychophysical experiments, which require accuracy in the presentation of stimuli and measurement of participants' response time.
The article is innovative, presented with scientific rigor and presents well -based arguments.
The experiment performed reproduces classic effects of audio-tactile paradigms confirming that the used smartphone has sufficient sensitivity to detect these phenomena. These results indicate a robust and suitable method for application in field research.
Technical and behavioral validation of experiments demonstrates potential to expand psychophysical and behavioral research to natural and remote contexts.
The results are presented clearly and associated with experimental needs.

Suggestions:
Although five models are a good starting point, variants of iOS devices could have been included, given the relevant presence of this type in experimental studies.
The sample of participants is limited, composed of a specific profile (young and university public). These results can be improved when expanding and varying the public profile in future work.
A deeper discussion about generalization limitations to the smartphone universe would also be interesting.

Then I suggest minor changes, including further discussion of general limitations of the approach in different operating systems/brands and inclusion of real scenarios of use by collecting data from different audiences.

Author Response

Thank you very much for taking the time to review this manuscript and for your constructive comments. Please find the detailed responses below and the corresponding revisions highlighted changes in the re-submitted file.

Comment 1: "Although five models are a good starting point, variants of iOS devices could have been included, given the relevant presence of this type in experimental studies."

Resonse 1: We agree with the reviewer that including only Android is a limitation. We have added a justification (page 11, line 355- 359) and a discussion of this point in the text (page 20 lines 610-618).

Comment 2: "The sample of participants is limited, composed of a specific profile (young and university public). These results can be improved when expanding and varying the public profile in future work."

Response 2: The sample of participants is indeed very homogeneous. It is however consistent with previous experiments done in our lab, which allows us to compare the results obtained on a smartphone with those obtained in a lab setting. We added a discussion of this point in the manuscript (page 20, lines 624-630).

Comment 3: "A deeper discussion about generalization limitations to the smartphone universe would also be interesting. I suggest minor changes, including further discussion of general limitations of the approach in different operating systems/brands and inclusion of real scenarios of use by collecting data from different audiences."

Response 3: We have added a paragraph about the limitations in the discussion session (page20, lines 608-633)

Reviewer 2 Report

Comments and Suggestions for Authors

The authors introduce a validation approach for assessing the precision perfomance of smartphones and demonstrate that  cognitive experiments requiring high precision in terms of audio and tactile characteristics can be successfully used by means of this kind of mobile devices. They establish a formal and well-defined procedure to measure the compliance of smartphones with the required thresholds of performance.

The work is worthwhile and provides significant contribution and should be published after some modifications, specified as follows:

- The structure of the paper is not typical. The authors should adhere to the typical structure of a scientific technical paper. For instance:

* The abstract should include work citations.

* The introduction must culminate with a paragraph that gives an overview regarding the organization of the rest of the manuscript.

* The authors should separate the discussion of related works is a separate section.

* When a given section is further divided into several subsection, the authors must provide a small introductory paragraph to describe what the reader should expect in that section.

* The paper does not have a conclusion section. A conlclusion section distinct form a discussion section.

- The authors must clearly show what is the mani contribution of this work relative to their own previous related works.

- It is not clear what is the purpose of Section 2. Are the procedures described therein of your proposal? Are they form previous work? Are they background foundation for your proposal? This needs to be explained.

- The authors must justify the selection of smartphones used in their assessement. Most commonly used smartphones should be considered and if not, the exclusion must be justified. For instance, the iPhone, which one of the most known worldwide, is not evaluated.

- Furthermore, the authors, should compare the accuracy of the proposed protocol and methodology to existing related procedures.

Author Response

We are grateful to the reviewer for their careful reading of our manuscript and for providing constructive feedback. Please find the detailed responses below and the corresponding revisions highlighted changes in the re-submitted file.

Comments 1: " The structure of the paper is not typical. The authors should adhere to the typical structure of a scientific technical paper. For instance:

* The abstract should include work citations.

* The introduction must culminate with a paragraph that gives an overview regarding the organization of the rest of the manuscript.

* The authors should separate the discussion of related works is a separate section.

* When a given section is further divided into several subsection, the authors must provide a small introductory paragraph to describe what the reader should expect in that section.

  • The paper does not have a conclusion section. A conclusion section distinct form a discussion section."

Response 1: This has been addressed: the manuscript now includes a final introductory paragraph describing the organization of the rest of the manuscript, a small introduction to each section and a conclusion section. We did however not separate the discussion of related work in a separate section as the introduction and justification of the present work necessitated to build on both the need expressed by the field of cognitive science and the gap in the existing related work. 

Comment 2: " The authors must clearly show what is the mani contribution of this work relative to their own previous related works."

Response 2: This has been made clearer in the manuscript (page 2, lines 67-74 & page 3, lines 99-103 & page 18, lines 547-558).

Comment 3: "It is not clear what is the purpose of Section 2. Are the procedures described therein of your proposal? Are they form previous work? Are they background foundation for your proposal? This needs to be explained."

Response 3: Section 2 originally combined three aspects: (1) the foundational stage of our work, which consisted in developing a methodology to assess smartphone hardware performance in delivering synchronized auditory and tactile stimuli; (2) the development of a method to enhance the precision of RT measurements; and (3) the application of both methods to assess five different smartphones. We agree with the reviewer that this presentation required clarification. We have therefore restructured the section and divided it into several parts with clearer titles.

Comment 4: "The authors must justify the selection of smartphones used in their assessement. Most commonly used smartphones should be considered and if not, the exclusion must be justified. For instance, the iPhone, which one of the most known worldwide, is not evaluated."

Response 4: We have added a justification for the selection of smartphones in the manuscript (page 3 lines 94-98 & page 11, lines 355-358) and a discussion of this point (page20, lines 610-620).

Comment 5: "Furthermore, the authors, should compare the accuracy of the proposed protocol and methodology to existing related procedures."

Response 5: This has been clarified in the manuscript, see page 1 lines 7-9 and page 18, lines 547-558.

Reviewer 3 Report

Comments and Suggestions for Authors

There are many key issues to be addressed before a recommendation of acceptance. Please consider the comments below.
Comment 1. Paper title:
(a) Since only five Android smartphone models were tested. The paper title should be updated to “Android Smartphones”.
(b) The experiment was about an audio-tactile paradigm; therefore, the paper title with “human behavior” is too general.

Comment 2. Abstract:
(a) References are not allowed. Therefore, update the abstract and the introduction.
(b) Research results (particularly numeric results) are missing.
(c) Discuss the research implications.

Comment 3. Add more terms in the “Keywords” to better reflect the scope of the paper.

Comment 4. The format in the list of references is not correct.

Comment 5. Carefully format the paper using in-text citations instead of footnotes if the content is more appropriate for in-text citations. For example, footnotes 1 and 2 should be changed to in-text citations.

Comment 6. Section 1 Introduction:
(a) Improper spacing is found in “(98 %)”.
(b) The style “Arthurs & al. (2021)” and “Nicosia & al. (2022)” is not appropriate. Please refer to the journal template.
(c) Significantly revise the literature review to focus mainly on the recent 5-year journal articles. The current version mainly covers references beyond the recent 5-year timeframe. In addition, ensure that the methodologies, results, and limitations of the existing works are discussed.
(d) Last paragraph: Strengthen the discussion of the research contributions with research elements and results.

Comment 7. Section 2 Assessment of hardware performance:
(a) Justify the selections of hardware.
(b) The “simplified pseudo-code” was presented in a Figure format, which is not appropriate.
(c) Elaborate on the details of Table 1.
(d) Include numbers for all equations.
(e) What if the frequencies of the touchscreen and accelerometer change?
(f) Table 2: It can be seen from the selected Smartphones that very limited versions of Android were tested.

Comment 8. A performance comparison between your method and existing works should be presented.

Comment 9. Please discuss how your method can be applied as a generic method, given that limited scenarios/settings were tested.

Author Response

Comments 1: "Paper title:
(a) Since only five Android smartphone models were tested. The paper title should be updated to “Android Smartphones”.
(b) The experiment was about an audio-tactile paradigm; therefore, the paper title with “human behavior” is too general."

Response 1: We agree and changed the paper title to "Using Android smartphones to collect precise measures of reaction times to multisensory stimuli".

Comments 2: "Abstract:
(a) References are not allowed. Therefore, update the abstract and the introduction.
(b) Research results (particularly numeric results) are missing.
(c) Discuss the research implications."

Response 2: we have updated the abstract according to the suggestions.

Comment 3: "Add more terms in the “Keywords” to better reflect the scope of the paper."

Response 3: We added more terms to better reflect the scope of the paper: "remote assessment; mobile application; timing precision; automated hardware testing ; reaction time; audio-tactile integration; peripersonal space. 

Comment 4: "The format in the list of references is not correct."

Response 4: the format has been corrected.

Comment 5: "Carefully format the paper using in-text citations instead of footnotes if the content is more appropriate for in-text citations. For example, footnotes 1 and 2 should be changed to in-text citations."

Response 5: This has been corrected.

Comments 6: "Section 1 Introduction:
(a) Improper spacing is found in “(98 %)”.
(b) The style “Arthurs & al. (2021)” and “Nicosia & al. (2022)” is not appropriate. Please refer to the journal template.
(c) Significantly revise the literature review to focus mainly on the recent 5-year journal articles. The current version mainly covers references beyond the recent 5-year timeframe. In addition, ensure that the methodologies, results, and limitations of the existing works are discussed.
(d) Last paragraph: Strengthen the discussion of the research contributions with research elements and results."

Response 6:

(a) and (b); these have been corrected.

(c) We added a very recent paper published this summer to the literature review. However, there are very few studies that address the issue of precise multisensory stimulus delivery using smartphones, and to our knowledge, no study has examined the precise measurement of reaction times to multisensory stimuli on these devices. Given the limited number of relevant papers, we included all references, even those older than five years. In the field of Cognitive Neuroscience, it is common for references to remain relevant over time, and citing older publications is therefore standard practice. Here also, we included all references, even those older than five years. 

(d) This has been done, see page 2, lines 75-93 and on page 3 lines 99-102. 

Comments 7: "Section 2 Assessment of hardware performance:
(a) Justify the selections of hardware.
(b) The “simplified pseudo-code” was presented in a Figure format, which is not appropriate.
(c) Elaborate on the details of Table 1.
(d) Include numbers for all equations.
(e) What if the frequencies of the touchscreen and accelerometer change?
(f) Table 2: It can be seen from the selected Smartphones that very limited versions of Android were tested.

Response 7:

(a) This has been done on page 3, lines 94-99 and on page 11 lines 355-359.

(b), (c),(d) These modifications have been made accordingly.

(e) This is now discussed on page 8, line 276-283.

(f) This limitation is now discussed on page 20 lines 610-620.

Comment 8: "A performance comparison between your method and existing works should be presented."

Response 8: we are not aware of any other work measuring reliability of reaction times to multisensory stimuli on smartphones. We highlighted the position of our study compared to existing works on page 2 lines 71-93, page 18 lines 547-557.

Comment 9: "Please discuss how your method can be applied as a generic method, given that limited scenarios/settings were tested."

Response 9: Our method to measure reaction time can be applied as a generic method, however, it is limited to android. This is discussed page 20 lines 618-620. 

Round 2

Reviewer 2 Report

Comments and Suggestions for Authors

The authors have correctly revised their manuscript addressing most of the concerns. However, some revisions were only partially completed.

1- The authors did provide a conclusion section, but it does not address the limitations of their proposal, nor does it offer ideas for future improvements. This is mandatory in any scientific technical paper.

2- Moreover, they did not separate related works into a distinct section, and the justification provided is not convincing. Additionally, the text has not been amended in response to the following comment:

"When a given section is further divided into several subsections, the authors must provide a brief introductory paragraph to describe what the reader should expect in that section."

However, they claim that they did. For instance, Section 2, Section 5, and Section 6.1 are such cases.

Author Response

Comment 1: The authors did provide a conclusion section, but it does not address the limitations of their proposal, nor does it offer ideas for future improvements. This is mandatory in any scientific technical paper.

Response 1: This has been added to the conclusion, see page 21 lines 670-676.

Comment 2: Moreover, they did not separate related works into a distinct section, and the justification provided is not convincing.

Response 2: We now modified the introduction section to present related works in a separate section, see page 3, lines 112-130.

Comment 3: Additionally, the text has not been amended in response to the following comment: "When a given section is further divided into several subsections, the authors must provide a brief introductory paragraph to describe what the reader should expect in that section." However, they claim that they did. For instance, Section 2, Section 5, and Section 6.1 are such cases

Response 3: Brief introductory paragraphs have been added: page 4, lines 133-133 ; page 9, lines 302-306 ; page 15, lines 436-437 and 439-442. 

Reviewer 3 Report

Comments and Suggestions for Authors

The quality of the paper has improved. However, some comments are being overlooked, and others remain inadequately addressed. I have some follow-up comments.
Follow-up Comment 1. In the abstract, the performance comparison between the proposed work and existing works should be summarized.
Follow-up Comment 2. Section 1 Introduction: The research contributions should be clearly stated.
Follow-up Comment 3. Figure 1:
- The labels of Channels 1 and 2 are missing.
- The caption did not specify the red trace.
Follow-up Comment 4. Figure 6: On the left-hand side, it is confusing that “UDP”, “ Audio”, “USB”, “Robotic finger”, “Computer”, “Audio interface”, “Phone”, and “Robotic finger” are not properly labelled.
Follow-up Comment 5. A performance comparison with the existing works should be discussed in the experiment section.

Author Response

Comment 1. In the abstract, the performance comparison between the proposed work and existing works should be summarized.

Response 1: To our knowledge, there is no work assessing smartphones temporal precision in both delivering multisensory audio-tactile stimulation and collecting reaction time. Thus, the performance comparison cannot be made. We modified the abstract to make this clearer. Moreover, we added that the RT methods we propose enhance the performance of the smartphones RT logging precision since the maximal precision with only touchscreen data is 8.33ms when the refresh rate of the device is maximal while we reach 4ms with our method.

Comment 2. Section 1 Introduction: The research contributions should be clearly stated.

Response 2: We added a clear statement of the research contributions page 2/3, lines 83-94.

Comment 3. Figure 1:
- The labels of Channels 1 and 2 are missing.
- The caption did not specify the red trace.

Response 3: Figure 1 was modified to correct this.

Comment 4. Figure 6: On the left-hand side, it is confusing that “UDP”, “ Audio”, “USB”, “Robotic finger”, “Computer”, “Audio interface”, “Phone”, and “Robotic finger” are not properly labelled.

Response 4: Figure 6 was modified to fix this.

Comment 5. A performance comparison with the existing works should be discussed in the experiment section.

Response 5: This has been added in the experiment section, see page 19, lines 567-574.